# Influence of Age and Body Size on the Dribbling Performance of Young Elite Soccer Players: A Cross-Sectional Descriptive Study

**DOI:** 10.3390/jfmk10020118

**Published:** 2025-03-31

**Authors:** Thiago V. Camata, Andrew H. Hunter, Nicholas M. A. Smith, Mathew S. Crowther, Marcelo Alves Costa, Felipe A. Moura, Robbie S. Wilson

**Affiliations:** 1School of the Environment, The University of Queensland, St Lucia, QLD 4072, Australia; tcamata@gmail.com (T.V.C.); a.hunter@uq.edu.au (A.H.H.); nma.smith@icloud.com (N.M.A.S.); 2School of Life and Environmental Sciences, The University of Sydney, Sydney, NSW 2000, Australia; mathew.crowther@sydney.edu.au; 3Sports Science, Centro Universitário Filadélfia, Londrina 86010-520, Brazil; 4Sport Sciences Department, State University of Londrina, Londrina 86057-970, Brazil; felipemoura@uel.br

**Keywords:** skill, performance, development, talent identification, football

## Abstract

Background/Objectives: Dribbling is a fundamental skill in soccer, but assessing the performance of youth players in this skill is complicated by the confounded effects of age and physical development. In this study, our aim was to quantify the interactive effects of age, height, and mass on the dribbling performance of 180 players between 10 and 21 years old from an elite Brazilian junior academy. Methods: For each player, we quantified their dribbling and sprinting speed along four different paths with varying curvature, and their ability to perform specific, directed dribbling drills using one or both feet. To characterise patterns of variation among player’s age, height, and mass—and to control for their confounding effects—we used a principal component analysis (PCA) to create a multivariate index of age and size (ASI). Results: Dribbling, sprinting, and directed dribbling were all positively associated with ASI; however, age alone was a better predictor of performance than ASI. Using multi-model inference, we found that a player’s overall dribbling was best predicted by models that included sprint speed and overall directed dribbling ability (*p* < 0.0001). When performing subsequent analyses that separate each of the directed dribbling drills into using dominant, non-dominant, or both feet, we found the best predictors of overall dribbling performance were sprinting and directed dribbling activities that use both feet. Conclusions: These results provide the first set of normative data for a detailed metric of dribbling performance and soccer-specific foot coordination that can allow players and coaches to compare and assess their performances relative to a single population of high-quality junior players

## 1. Introduction

Soccer is the world’s most popular team sport and dribbling is one of the most important attacking strategies underlying an individual’s success [1,2,3]. The ability to take on and beat defenders creates scoring opportunities, opens spaces for team-mates, and drags defenders out of their preferred positions [4,5,6]. Those individuals that are most effective at this skill are often the most admired and expensive players in professional football, and clubs often seek to recruit the top players in this skill they can afford [6,7,8]. Development of dribbling ability is considered a pivotal attribute in the development of young players [9,10,11,12,13,14,15,16,17], and identifying junior players that are accomplished dribblers and nurturing this capacity is critical to the long-term financial success of any football academy [18]. Recent improvements in measuring dribbling abilities now offers the opportunity for junior football academies to quantify the performances of large numbers of individuals and compare them to others of their age and development stage [14,19,20]. However, to facilitate such an endeavour, one requires the collection of normative data from talented junior players with reference to their age, height, and mass so that players can be equitably compared with others in their age cohort [14].

Dribbling in matches is a complex, multi-player action that relies on the perceptual, cognitive, and motor skills of both the attacker and the defender [3,6,15,16,17,19,21]. Players can approach and beat opponents in limited spaces using close control of the ball and rapid changes in direction, or in open and wide positions using both close control and high sprint speeds to beat opponents. Thus, when designing a metric of individual dribbling performance, one must consider both the complexities and varying styles of this skill and demonstrate that the metric is associated with a player’s actual performances in match-relevant scenarios [3,16,19,22,23]. Recently, we developed a measure of dribbling performance that quantifies an individual’s ability to dribble the ball when moving along different paths of varying curvature [20], allowing one to capture both an individual’s ability to move at high speeds in straight lines and along curved paths that require closer control and rapid changes in direction [20,24]. Furthermore, previous studies have shown this metric of dribbling performance is reliable and repeatable and significantly predicts individual variation in 1v1 attacking performance [24], goal scoring success in training games [25], defending success in 1v1 games [20], and overall success in 3v3 games [26]. Taken together, by collecting normative data using a measure of dribbling performance [20] one can provide a robust and match-relevant metric of performance that allows individuals to be compared within teams, among age-groups, competitions, and nations. Because dribbling performance will be affected by a player’s age [12,27], height [27,28], and mass [27], understanding how these three traits interact will also allow one to statistically control for these variables when making comparisons among individuals [29].

Identifying those technical and athletic traits that underlie dribbling performances can also allow coaches and skill acquisition specialists to develop targeted training regimes for junior player development [6,16,20,23]. Surprisingly, the relative importance of sprinting speed and foot coordination for dribbling performance has not been previously investigated but could elucidate the mechanisms supporting higher performances [20]. For example, it is unclear whether high dribbling speeds require only high technical control with the dominant foot rather than both feet. One could intuitively speculate that the best predictor of dribbling performance would be coordination with the dominant foot, given some of the world’s best dribblers (e.g., Messi, Maradona) seem to rarely touch the ball with their non-dominant foot. However, high technical control with both feet may allow players to have more options available to them than players who are “one-footed, which is in line with Gibson’s theory of affordances [30,31,32]. Thus, high technical control with both feet may allow one to react faster when small mistakes are made, or when rapid changes in direction are required. Regardless of the underlying mechanism, determining the best predictors of dribbling speed will allow more targeted practice regimes for young players.

The aim in this study was to quantify the interactive effects of age, height, and mass on the dribbling performance of players from an elite Brazilian junior academy. To do this, we quantified the dribbling and sprinting speed of each player along four different paths of varying curvature [20].These results provide the first set of normative data for a detailed metric of dribbling performance that will allow coaches to control for indices of age and body size and compare their players against a population of high-quality junior players. Describing the normative functions for dribbling speed is a necessary first step for developing a robust quantitative methodology for assessing players within age-cohorts that is both objective and equitable [16]. Finally, we also aimed to describe the underlying determinants of dribbling performance by testing each individual’s football-specific coordination when performing one-footed dribbling drills (dominant or non-dominant) and two-footed drills. We predicted that for this population of elite junior players, age, size, sprint speed, and foot coordination would all be significant predictors of dribbling speed.

## 2. Methods

### 2.1. Study Design

We measured the dribbling and sprinting performance of all 180 individuals across seven teams (U12, U13, U14, U15, U16, U17, U20) from a Tier 1 professional academy that compete in their state and national competitions in Brazil. For each individual, we also recorded their date of birth, mass (±0.1 kg; Ulysses, USA), and standing height (±0.01 m). All players and parental and legal guardians gave consent to be involved in the study, which was in accordance with ethical protocols for the University of Queensland, Australia, and State University of Londrina, Brazil (#2019001398). All data were analysed anonymously. No activities or movements beyond those regularly utilised by the players in their training sessions were involved in the study design.

In November 2019, each age group attended two, two-hour sessions, with one day of rest between sessions. Prior to each session, players proceeded through their normal 15 min warm-up routine with their coaches. Depending on the number of individuals in their specific training squad, players were split into groups of three or four, with groups rotating through each testing station. All groups progressed through stations in the same sequence, but because each group was randomly assigned an initial station, the order of testing differed among them. On day 1, we measured dribbling and sprinting performance in the following sequence: dribbling/sprinting on Path 1, dribbling/sprinting on Path 4, dribbling/sprinting on Path 3, and dribbling/sprinting on Path 2. On day 2, we measured directed dribbling in the following sequence: left foot dribble test 1, right foot dribble test 1, right foot dribble test 2, left foot dribble test 2, both feet dribble test 1, and both feet dribble test 2. Players were allowed a brief period to familiarise themselves with each task.

### 2.2. Dribbling and Sprinting Performance Along Curved Paths

The dribbling and sprinting performance of each player was measured along four different 30 m long paths that varied in curvature [20]. Each path consisted of a 1 m-wide channel with outer boundaries marked with a 6 mm black and yellow plastic chain (Kateli, Brazil). Paths consisted of a series of straight sections that were interspersed with turns that were always 1 m in diameter and either 45° (1/8 of a circle), 90° (1/4 of a circle), 135° (3/8 of a circle), or 180° (1/2 a circle). Path 1 had no turns and consisted of a straight 30 m long path. Path 2 had 6 turns, with five 90° turns and one 135° turn, with a total curvature of 0.37 radians.m^−1^. Path 3 had 10 turns, with three 45° turns, two 90° turns, two 135° turns, and three 180° turns, with a total curvature of 0.67 radians.m^−1^. Path 4 had 15 turns, with four 45° turns, four 90° turns, two 135° turns, and five 180° turns, with a total curvature of 1.03 radians.m^−1^.

To record the time taken to dribble/sprint along each path, light gates (fusion sport, SMARTSPEED PT 2 Gate System), with the beam positioned at waist height, were placed at the entrance and exit to each path. On each path, players started 0.5 m before the entrance and either dribbled a size 5 ball or sprinted along the path as fast as possible. When sprinting, players were required to keep their entire body within the boundary of the chains as they navigated through the path. When dribbling, the ball was required to stay within the chains, but the player’s body did not. If players cut corners, the test was stopped, and the trial repeated after a minimum rest of 30 s. To quantify dribbling and sprinting speed, we divided the distance (30 m) by the time taken to complete a trial to calculate an average speed per trial. Players completed three dribbling trials and two sprinting trials on each path. For each path, the average of the dribbling trials was the measure of dribbling performance, and the average of the sprinting trials was the measure of sprinting performance. Thus, for each player there was one measure of sprinting and dribbling performance along each of the four paths.

We used separate principal component analysis (PCA) to characterise patterns of variation among our correlated measures, creating overall measures of dribbling (PC_D_) and sprinting (PC_S_) performance. The first component of the PCA on dribbling (PC_D1_) explained 79.9% of the variation observed in the data (Appendix Table A1). All data were standardised (0 ± 1) prior to these analyses. Because all vectors of PC_D1_ loaded in the same direction, and larger positive values were indicative of higher dribbling speeds over all paths, PC_D1_ represented overall dribbling performance. The second component of the PCA of dribbling (PC_D2_) only explained 9.3% of the variation (Appendix Table A1). The first component of the PCA on sprinting (PC_S1_) explained 86.7% of the variation observed in the data (Appendix Table A2). Because all vectors of PC_S1_ loaded in the same direction, and larger positive values were indicative of higher sprinting speed over all paths, PC_S1_ represented overall sprinting performance. The second component of the PCA of sprinting (PC_S2_) only explained 6.6% of the variation (Appendix Table A2).

### 2.3. Directed Dribbling

The players’ ability to execute specific dribbling skills was assessed with six tasks, two using only the left foot, two using only the right foot, and two using both feet. Players self-reported which foot was their dominant foot. Each task required players to dribble a size 5 ball through 15 cones spaced 0.8 m apart (12 m total linear distance) executing a specific skill between each cone. Players were instructed to complete the task as quickly as possible without making a mistake. Using light gates (fusion sport, SMARTSPEED PT 2 Gate System), the time to complete the task was measured (raw time), along with the number of times the skill was not perfectly executed (errors). Regardless of the magnitude of the mistake, a maximum of 1 error could be attributed between any two cones. For example, if a player took one extra (or fewer) touch than specified, this was scored as 1 error. If a player took three extra touches between two cones, this was still scored as 1 error. To account for errors, each players’ time to complete the task was adjusted with the formula $adjustedtime\left(s\right)=rawtime+\left(0.1*errors\right)$. The adjusted time was then used to calculate the average dribbling speed (m.s^−1^ = $\frac{12}{adjustedtime}$) for each trial.

Players executed each dribbling task three times and for each task the average of their trials was their measure of performance. Prior to analysis, dribbling tasks with just the left foot and just the right foot were converted to the dominant foot and nondominant foot for each player. We used a PCA to characterise variation among our measures of directed dribbling. The first component (PC_DD1_) of a PCA of directed dribbling explained 70.8% of the variance in the data (Appendix Table A3) and was our metric of overall direct dribbling performance. The first component (PC_DF1_) of a PCA on data from the directed dribbling tests using only the dominant foot explained 86.9% of the variance in the data (Appendix Table A4) and was our metric of dominant foot dribbling. The first component (PC_NDF1_) of a PCA on data from the directed dribbling tests using only the nondominant foot explained 87.5% of the variance in the data (Appendix Table A5) and was our metric of nondominant foot dribbling. The first component (PC_BF1_) of a PCA on data from the directed dribbling tests using both feet explained 82.2% of the variance (Appendix Table A6). For all PCA of directed dribbling, all vectors loaded in the same direction with larger positive values, indicating faster speeds.

### 2.4. Directed Dribbling Tasks

Right foot test 1: Players performed one touch between each cone. The first touch used the lateral part of the right foot (outside of the foot) to take the ball to the right side of the first cone, then using the medial part of the right foot (inside of the foot) players moved the ball to the left side of the next cone. This sequence of touches was repeated until the final cone was passed.

Left foot test 1: Identical to Technique #1 but using the left foot.

Right foot test 2: Players performed one touch between each cone. The first touch used the lateral part of the right foot to take the ball to the right side of the first cone, then, using the underside of the right foot, players moved the ball to the left side of the next cone. This sequence of touches was repeated until the final cone was passed.

Left foot test 2: Identical to Technique #3 but using the left foot.

Both feet test 1: The player performed two touches between each cone. Initially, the first touch used the lateral part of the right foot to take the ball to the right side of the first cone, then, using the medial part of the right foot, the player moved the ball towards the left side of the next cone. After which, the player used the lateral part of the left foot to take the ball to the left side of the second cone, then, using the medial part of the left foot, the player moved the ball back towards the right side of the next cone. This sequence of touches was repeated by the player until the final cone was passed.

Both feet test 2: The player performed two touches and a step-over between each cone. The first two touches used the lateral of the right foot to take the ball towards the right side of the first cone, then, the player performed a step-over (from left to right) with the right leg, then, using the lateral part of the left foot, the player moved the ball back towards the left side of the next cone. The player then performed another touch using the lateral part of the left foot to take the ball towards the left side of the second cone then, used the left leg to do a step-over the ball (from right to left), and then, using the lateral part of the right foot, the player cut the ball back towards the right side of the next cone. This sequence of touches was repeated until the final cone was reached.

### 2.5. Statistical Analysis

We used principal component analysis (PCA) to characterise patterns of variation among player’s height, mass, and age (birth date was converted to decimal age), creating a multivariate age and size index. The first component (PC_ASI_) accounted for 89.4% of the variance in the data (Appendix Table A7) and all vectors loaded in the same direction with larger positive values indicating greater age and size.

For descriptive summary statistics, separate one-way ANOVA and Tukey’s Honest Significant Difference (95% confidence intervals) [33] were used to detect significant differences between teams for dribbling speed (PC_D1_), sprinting speed (PC_S1_), and directed dribbling speed (PC_DD1_). Prior to analysis, a square transformation was used to normalise the left-skewed directed dribbling (PC_DD1_) data. To estimate the relationships between our anthropometric measures and each measured trait, separate linear models [33] were used to determine the effects of age, height, mass, and age and size index (ASI) on dribbling speed (PC_D1_), sprinting speed (PC_S1_), and directed dribbling (PC_DD1_).

To determine the underlying predictors of dribbling performance on the curved paths, we used a linear model [33] to estimate the effects of age and size index (ASI), directed dribbling (PC_DD1_), and sprinting (PC_S1_) on dribbling (PC_D1_). Prior to analysis, all predictor variables were rescaled to a mean of 0 and standard deviation of 1 to better compare their relative effect and were tested for normality. Normality of residuals was assessed using the Shapiro–Wilk test and visual inspections of Q-Q plots using R. We used power analyses to determine if we had sufficient statistical power to detect significant effects for both continuous traits (regression) and among groups (ANOVA) [33]. We assumed a medium effect size in each case, an alpha level of 0.05, and power of 0.80. Results indicated we had sufficient statistical power across all analyses. We estimated the most likely effect of each variable using multi-model inference based on information theory [34]. Initially, we estimated parameters in the full model, which included all main effects and two-way interactions. Then, using the MuMIn library of R [33,35] we estimated the parameters of all possible sub-models, including the null. For each model, this method generates an Akaike weight, which describes the likelihood that model explains the data better than all other models. With these Akaike weights, we calculated a weighted average for each parameter included in our models. Unlike null hypothesis testing, this method accurately estimates the most likely effects of variables as all possible models (including the null) contribute to the value of each parameter.

We also tested if dribbling performance along paths was constrained by directed dribbling with a specific foot (dominant or nondominant). To test this, we split directed dribbling into its three components: dominant foot dribbling (PC_DF1_), nondominant foot dribbling (PC_NDF1_), and both feet dribbling (PC_BF1_). Then, with multi-model inference, we used a linear model to estimate the effects of age and size index (ASI), dominant foot dribbling (PC_DF1_), nondominant foot dribbling (PC_NDF1_), both feet dribbling (PC_BF1_), and sprinting (PC_S1_) on dribbling performance along the curved paths (PC_D1_).

## 3. Results

Each player’s age (14.65 ± 2.5 years; range = 10.1–20.5 years), mass (59.3 ± 15.11 kg, range = 31.7–93.0 kg), and height (1.68 ± 0.14 m, range = 1.35–1.96 m) were recorded on the first day of assessment.

### 3.1. Sprint Performance

Sprinting performance (PC_S1_) significantly differed across age groups, increasing from a mean of −2.03 ± 2.18 for the U12 team, to a mean of 2.64 ± 1.56 for the U20 team (Table 1). Sprinting performance was positively associated with age (R^2^ = 0.426, t = 11.27, *p* < 0.001), height (R^2^ = 0.273, t = 8.00, *p* < 0.001), mass (R^2^ = 0.278, t = 8.10, *p* < 0.001), and age and size index (ASI) (R^2^ = 0.358, t = 9.73, *p* < 0.001). The linear relationships between sprinting performance and age, mass, height, and ASI are provided in Table 2.

### 3.2. Directed Dribbling Performance

While directed dribbling performance (PC_DD1_) tended to increase from the U12 team (−1.70 ±2.20) to the U20 team (1.06 ±1.420), only the U12 and U14 teams statistically differed from other teams (Table 1). Directed dribbling performance (PC_DD1_) was significantly positively associated with age (R^2^ = 0.133, t = 5.01, *p* < 0.001), but not with height (R^2^ = 0.011, t = 1.36, *p* = 0.173) and mass (R^2^ = 0.015, t = 1.60, *p* = 0.111) (Figure 1). A player’s ASI was significantly positively associated with directed dribbling performance (R^2^ = 0.042, t = 2.68, *p* < 0.01) (Figure 2). The linear relationships between directed dribbling performance and age, mass, height, and ASI are provided in Table 2.

### 3.3. Dribbling Performance

Dribbling performance (PC_D1_) significantly differed across teams, increasing from a mean of −1.73 ±1.35 for the U12 team to a mean of 1.88 ±1.19 for the U20 team (Table 1). Dribbling performance (PC_D1_) was significantly positively associated with age (R^2^ = 0.398, t = 10.42, *p* < 0.001), height (R^2^ = 0.127, t = 4.88, *p* < 0.001), and mass (R^2^ = 0.138, t = 5.11, *p* < 0.001) (Figure 1). A player’s ASI was also significantly positively associated with dribbling performance (R^2^ = 0.225, t = 6.88, *p* < 0.001) (Figure 2). The linear relationships between dribbling performance and age, mass, height, and ASI are provided in Table 2.

Table 3 shows the parameter estimates of our statistical models of dribbling performance estimated by multi-model inference (Table A8). These estimates reveal those factors affecting variation in overall dribbling performance along the curved paths. Individuals with faster sprinting performances were more likely to have better dribbling performances on the curved paths. Additionally, individuals with better directed dribbling performance were likely to have better performance on the dribbling paths. When directed dribbling was split into its three components, only sprinting and dribbling performance with both feet were associated with dribbling performance along paths (Table 4 and Table A9).

## 4. Discussion

In this study, we assessed how a player’s sprinting performance, directed dribbling, and age and size index (age, height, and mass), affects their overall dribbling performance. Dribbling performance on the curved paths was best predicted by a player’s directed dribbling and sprinting speed. When we separated directed dribbling into dribbling with dominant, nondominant and both feet, we found the best predictors for dribbling performance to be a player’s directed dribbling with both feet and sprinting performance. Our results support previous studies that also found a player’s overall dribbling performance was associated with their age [12,27], height [27,28] and mass [27], meaning older, larger players tend to be better at dribbling. A player’s overall sprinting performance (PC_SP1_) was also associated with their age, height and mass. Further, dribbling (PCD1), sprinting (PC_SP1_) and directed dribbling (PC_DD1_) were positively associated with a player’s age and size index (ASI). 

Age may better predict dribbling and sprinting than the combined metric of ASI because for any given age, shorter players may have an advantage due to a lower centre of gravity and increased agility. To further explore this idea, we regressed height on age to estimate residual height for each player. Then, separately for dribbling (PC_D1_), sprinting (PC_SP1_), and directed dribbling (PC_DD1_), we ran a linear model to estimate the relationship between residual height and each trait. We found a weak negative relationship for dribbling (PC_D1_) (β = −0.037, *p* = 0.01, R^2^ = 0.03) and directed dribbling (PC_DD1_) (β = −0.059, *p* = <0.001, R^2^ = 0.07), but not sprinting (β = −0.008, *p* = 0.719, R^2^ = −0.005), suggesting shorter players tend to have better dribbling performances than taller players. Previous work by Malina et.al. [28] also report a negative correlation between a player’s height and a composite score of technical skill based on six soccer-specific tests that included two measures of dribbling speed. When calculating the relationship between ASI and dribbling/sprinting performance, we assumed that age, height, and mass all independently affect each trait in a positive direction. Because age and height may affect dribbling and sprinting in different directions, one of our initial assumptions for calculating the ASI may not be appropriate. Although age may be a better predictor of dribbling and sprinting performance than ASI, this is not the case for other technical skill traits. For example, ASI was a better predictor of kicking speed than age, height, or mass alone [29]. This suggests the effect of biological development on performance is not the same for all technical skill traits. Such differences are not entirely surprising, since traits such as kicking speed rely heavily on muscle power while others rely more on technique, with physical development offering little to no advantage. Therefore, in the context of talent identification and controlling for developmental bias when assessing skills, coaches and researchers need to first identify which aspects of development (age alone, height alone, mass alone, or ASI) best predict their target trait. To control for the effects of age when identifying talented youth players, coaches can use the equations we developed in Table 2. Using dribbling performance along the curved paths as an example, we can estimate a player’s expected dribbling performance with dribbling = −6.449 + (0.436 × age), then compare that against their actual performance to identify which players are performing better or worse than expected for their age.

Both a player’s sprinting speeds and their directed dribbling abilities equally predicted dribbling speed along the curved paths. To determine which traits best predicted dribbling performance, we also separated directed dribbling into tests of dominant foot, nondominant foot, and dribbling with both feet. We found the best predictors of a player’s overall dribbling performance to be their directed dribbling with both feet, and sprinting. Most soccer players, both amateur and professional, favour using their dominant foot for all actions during games [36,37], with some of the world’s best players using their dominant foot almost exclusively (e.g., Lionel Messi, Diego Maradona). From this, one may conclude proficiency with the nondominant foot is not essential to be a successful soccer player, but our results suggest otherwise. Furthermore, the metric of dribbling performance used in this test also predicts a player’s ability to successfully dribble passed defenders in 1v1 game-like situations [24], giving further support to the idea that players with greater proficiency with both feet may be at a competitive advantage than those with just high performance with their dominant foot. Better dribbling performances in players with high technical ability with both feet is consistent with Gibson’s theory of affordances [30,31,32]. In this case, affordance theory predicts that players who can dribble effectively with both feet have more options available to them than players who are “one-footed”. For example, even when dribbling along curved paths, using the nondominant foot to change the direction of the ball may be the most efficient (fastest) option available. A player that can use both feet is more likely to both perceive the benefits of using their nondominant foot and use it accordingly when it offers the faster option. Conversely, a one-footed player may only perceive using their dominant foot as an option and execute this action despite it being the slower option. 

There is little doubt that possessing close ball control in isolated conditions will be a pre-requisite for match-realistic dribbling ability against opponents, but there is no guarantee that training-induced increases in technical dribbling will directly map onto attacking performance against opponents. This is a key limitation of our study. Dribbling speeds, as measured along the curved paths, are only correlated with an individual’s attacking and defensive ability in 1v1 competitions [20,24,25] and we do not know the causal relationship between our metric of dribbling and the actual activity utilised in game-realistic situations. Our measure of dribbling performance relies on the execution of closed skills, which have often been criticised for de-contextualising skills from the performance setting [38,39,40]. However, we do expect that improvements in dribbling speed along the testing paths will be associated with improvements in a player’s attacking and defensive capabilities during match-realistic conditions, but this still remains to be tested.

Using data to augment selection practices is now common among elite professional clubs but it is unfortunately still rare at youth levels [41,42]. The opinion of expert scouts who observe and evaluate potential recruits during matches and training sessions are still the dominant method of identifying talented youth players [27,41,42,43]. Scouts must often assess players from just single matches or sessions but because youth players compete within annual age cohorts there is still substantial variation in age and physical maturity within each group that complicates the missions of scouts. For example, players born in the first month within the annual cohort year of under 13s competitions will be approximately 10% older than those born in the last month. Because older players are more likely to be judged as better because they are likely to be physically stronger, youth football academies across the world are dominated by individuals born in the first half of the year [44]. This selection bias—referred to as the relative age-effect—is one of the most insipid forms of discrimination in team sports. By comparing players using metrics of performance that are corrected for age and size, one could then provide fairer, more robust, and accurate assessments of relative performances. In this study, we provide normative data that can allow one to assess the soccer-specific skill of youth soccer players after being adjusted for age and size. With the normative equations provided, one can more equitably compare players’ dribbling performances. However, one should be mindful that these equations are based on athletes from only one group of elite youth players, and it remains to be tested how generalizable these are to players from other clubs, countries, or continents. Regardless, the selection of talented players should be rapidly moving towards the inclusion of quantitative metrics that help minimise biases of age and size—such as the metrics described in this study—to ensure both selection fairness and accuracy in youth team sports.

## Figures and Tables

**Figure 1 jfmk-10-00118-f001:**
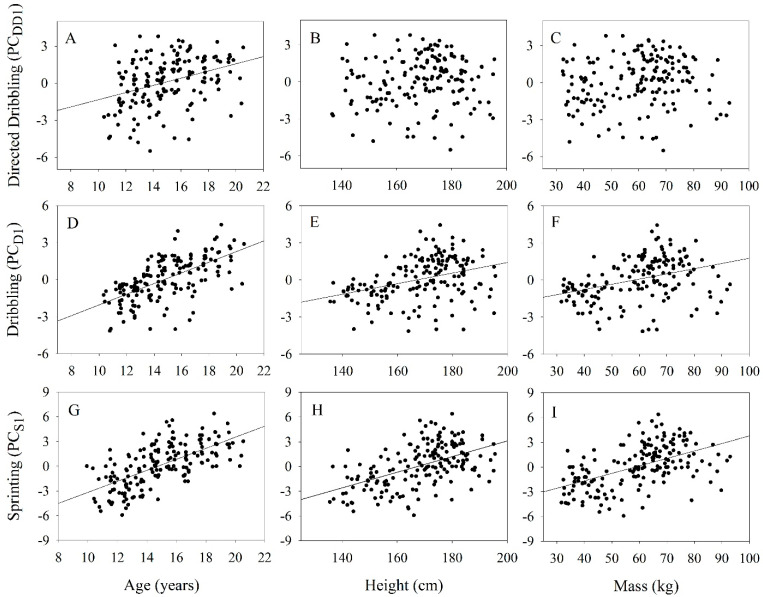
The relationships between age, height, and mass with an individual’s overall directed dribbling performance (PC_DD1_) (**A**–**C**, respectively), overall dribbling performance (PC_D1_) (**D**–**F**, respectively), and overall sprinting performance (PC_S1_) (**G**–**I**, respectively). Regression lines indicate significant relationships (*p* < 0.05).

**Figure 2 jfmk-10-00118-f002:**
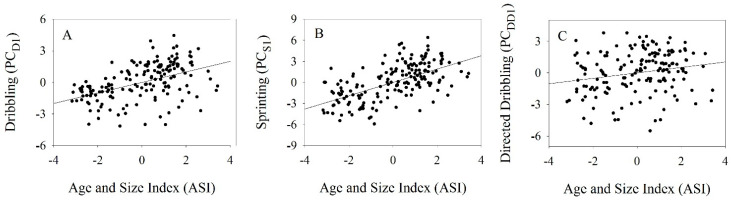
The relationships between age and size index (ASI—based on the first component of a PCA on age, height, and mass) with (**A**) overall dribbling performance (PC_D1_), (**B**) overall sprinting performance (PC_S1_), and (**C**) overall directed dribbling performance (PC_DD1_). Regression lines indicate significant relationships (*p* < 0.05).

**Table 1 jfmk-10-00118-t001:** Overall sprinting, dribbling, and directed dribbling performances for each team. Mean and standard deviation values are presented. Group differences were identified with ANOVA and Tukey multiple comparisons.

		Overall Performances	
Team	Sprinting (PC_S1_)	Dribbling (PC_D1_)	Directed Dribbling (PC_DD1_)
U12	−2.03 ± 2.18 (*n* = 30)	−1.73 ± 1.35 (*n* = 25)	−1.70 ± 2.20 (*n* = 25)
U13	−2.53 ± 1.86 (*n* = 22)	−1.24 ± 0.94 (*n* = 22)	−0.36 ± 1.87 (*n* = 21)
U14	−1.49 ± 2.20 (*n* = 25)	−0.74 ± 1.41 (*n* = 22)	−0.59 ± 2.31 (*n* = 24)
U15	0.21 ± 2.00 (*n* = 24)	0.20 ± 1.39 (*n* = 23)	−0.07 ± 1.63 (*n* = 23)
U16	2.06 ± 1.86 (*n* = 22)	0.80 ± 1.80 (*n* = 20)	0.65 ± 1.78 (*n* = 21)
U17	2.64 ± 1.48 (*n* = 29)	1.34 ± 1.36 (*n* = 29)	0.71 ± 1.92 (*n* = 29)
U20	2.64 ± 1.56 (*n* = 32)	1.88 ± 1.19 (*n* = 30)	1.06 ± 1.42 (*n* = 30)

Sprinting: Significant differences (*p* < 0.001) between U12: U15, U16, U17, U20; U13: U15, U16, U17, U20; U14: U16, U17, U20; U15: U20; U17: U20. Dribbling: Significant differences (*p* < 0.01) between U12: U15, U16, U17, U20; U13: U15, U16, U17, U20; U14: U16, U20; U15: U20; U17: U20. Directed Dribbling: Significant differences (*p* < 0.01) between: U12: U16, U17, U20; Significant differences (*p* < 0.05) between: U14: U20.

**Table 2 jfmk-10-00118-t002:** Effects of age, height, mass, and age and size index (ASI) on dribbling and sprinting along paths and directed dribbling. Each pairing of intercept and β represents a separate linear model.

	Dribbling (PC_D1_)		Directed Dribbling (PC_DD1_)		Sprinting (PC_S1_)	
	Intercept	β	Intercept	β	Intercept	β
Age	−6.449 *	0.436 *	−4.301 *	0.29 *	−9.911 *	0.671 *
Height	−7.160 *	0.042 *	−2.480	0.015	−15.895 *	0.095 *
Mass	−2.448 *	0.042 *	−0.902	0.016	−5.288 *	0.090 *
ASI	0.057	0.499 *	0.082	0.257 ^+^	0.091	0.948 *

* *p* < 0.001; + *p* < 0.01.

**Table 3 jfmk-10-00118-t003:** Parameter estimates for model of dribbling performance along curved paths. For each variable, a weighted average of the parameter value from all models was calculated using Akaike weights. Predictor variables were rescaled to a Mean of 0 and SD of 1 to better compare their relative effect on dribbling performance. ASI: Age and size index.

Parameter	Estimate	SE	z	*p*	Importance
Intercept	−0.081	0.091	0.888	0.375	
ASI	0.067	0.056	1.199	0.230	0.96
Directed dribbling	0.366	0.042	8.586	<0.0001	1
Sprinting	0.348	0.038	8.974	<0.0001	1
ASI: Directed dribbling	0.055	0.036	1.492	0.136	0.82
ASI: Sprinting	0.018	0.024	0.743	0.457	0.51
Directed dribbling: Sprinting	−0.006	0.012	0.515	0.606	0.40

**Table 4 jfmk-10-00118-t004:** Parameter estimates for model of dribbling performance along curved paths with directed dribbling split into its three components (dominant foot, nondominant foot, both feet). For each variable, a weighted average of the parameter value from all models was calculated using Akaike weights. Predictor variables were rescaled to a Mean of 0 and SD of 1 to better compare their relative effects on dribbling performance. ASI: Age and size index.

Parameter	Estimate	SE	z	*p*	Importance
Intercept	0.028	0.107	0.265	0.790	
Directed dribbling (both feet)	0.375	0.103	3.587	<0.001	1
Directed dribbling (dominant foot)	0.160	0.126	1.257	0.208	0.85
ASI	0.012	0.091	0.134	0.893	0.78
Sprinting	0.323	0.049	6.411	<0.001	1
Directed dribbling (both feet): ASI	0.076	0.097	0.783	0.433	0.50
Directed dribbling (non-dominant foot)	0.107	0.114	0.926	0.354	0.72
Directed dribbling (both feet): Directed dribbling (dominant foot)	−0.011	0.039	0.283	0.777	0.24
Directed dribbling (dominant foot): ASI	0.021	0.062	0.341	0.732	0.23
ASI: Sprinting	−0.004	0.022	0.188	0.851	0.19
Directed dribbling (dominant foot): Sprinting	−0.003	0.021	0.156	0.875	0.21
Directed dribbling (dominant foot): Directed dribbling (non-dominant foot)	−0.007	0.038	0.187	0.851	0.16
Directed dribbling (both feet): Sprinting	0.003	0.021	0.160	0.872	0.25
Directed dribbling (both feet): Directed dribbling (non-dominant foot)	−0.002	0.029	0.078	0.937	0.18
Directed dribbling (non-dominant foot): Sprinting	−0.001	0.018	0.061	0.951	0.17
ASI: Directed dribbling (non-dominant foot)	0.009	0.043	0.214	0.830	0.17

## Data Availability

Data are available upon request from the corresponding author.

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
