# Peer review of "Influence of Age and Body Size on the Dribbling Performance of Young Elite Soccer Players: A Cross-Sectional Descriptive Study"

_jfmk, 2025, doi:10.3390/jfmk10020118_

Round 1
Reviewer 1 Report
Comments and Suggestions for Authors
The manuscript is entitled “Effects of age and body size on dribbling performance of elite junior soccer players.". The article raises some interesting questions but needs some adjustments before it can be published. Here are our contributions. Thank you for the opportunity to read the manuscript

There is a need to revise the English-language work in its entirety.
Author Response
Thank you for your extensive review and comments. Below I have replied to each of the comments/suggestions - I appreciate the time and effort that has gone into the reviews.
Reviewer 1
The following are suggestions for adjustments and possible issues to be considered by the authors:
- Title: Effects of age and body size on dribbling performance of elite junior soccer players
Suggestion: Influence of morphometric indicators on the dribbling performance of young soccer players: a cross-sectional descriptive study
REPLY: We have changed the title to reflect the reviewer’s suggestion:
“Influence of age and body size on the dribbling performance of young elite soccer players: a cross-sectional descriptive study”
Abstract
- We suggest making the changes in the abstract if the other suggestions are accepted in the other sections of the manuscript.
- We suggest inserting the justification or importance of the study.
- The aim should be concise and clear e.g. “The aim of this study was.....”
- The following is a structured evaluation of the abstract, with objective points for improvement:
- It is suggested that the method be rewritten with the study design; description of participants and variables investigated, citation of tests and instruments, and general type of statistical analysis.
- Present the main results concisely and their (p≤). Relate the findings to applications in the sports field.
- Be incisive in your conclusion
REPLY: We have now endeavoured to achieve these stated suggestions in the Abstract. Thank you for your clear advice and suggestions.
Keywords
Avoid repeating the words and terms mentioned in the title.
REPLY: Corrected
Introduction
- We strongly suggest updating the references used in the introduction, looking for articles from the last 5 years.
REPLY: Indeed, this is important. We have now added in appropriate references from the last five years.
- We suggest an introductory paragraph on the concept of soccer, the motor characteristics that are developed and the number of male soccer players. Fundamentally the object of study of the work.
REPLY: We have tried to be more explicit about the objective of the study and now mentioned some of the concepts of soccer.
- Lack of reference in the paragraph: “The ability to take on and beat defenders can create goal-scoring opportunities, open spaces for teammates and drag defenders out of their preferred positions. The individuals who are most effective in this capacity are often the most admired and expensive players in professional soccer, and clubs often seek to recruit the best players in this capacity that they can afford.” It is suggested to insert references from the last 5 years strongly.
REPLY: We have now added appropriate references here.
- Lack of reference in paragraph “Recent improvements in the measurement of dribbling skills now offer junior soccer academies the opportunity to quantify the performances of a large number of individuals and compare them with others of their age and stage of development.” Strongly suggest inserting. Describe the measurement methods used.
REPLY: We have now added in citations.
- Lack of reference in paragraph: “However, to facilitate this effort, it is necessary to collect normative data from talented junior players with reference to their age, height and mass, so that players can be compared with others at a similar stage of development. Currently, no such data exists for dribbling skills.” It is strongly suggested to insert references from the last 5 years.
REPLY: We have now added in citations.
- Lack of reference in paragraph: “Players can approach and beat opponents with tight ball control and quick changes of direction in tight positions, while in open or wide positions players can use both control and running speed to beat opponents. Designing a dribbling performance metric that is a 'closed skill' requires considering both the complexities and varied styles of this skill and demonstrating that the metric is associated with performances in game-relevant scenarios.” It is advisable to insert references from the last 5 years.
REPLY: We have now added in citations.
- Clearly and objectively describe the protocol developed by Wilson et al. (2018). Which measurement instrument?
REPLY: The text now reads: “Because the methodology uses paths of different curvature, it can capture an individual’s ability to move at high speed in a straight line where few touches and high sprint speeds are required, and along paths that require closer control and rapid changes of direction. An overall measure of dribbling performance can then be taken from the first dimension of a principal component analysis on performances across all paths. This protocol provides a highly robust metric of dribbling that includes between 9-15 individual measures of dribbling performance.”
- About the paragraph: “Biological maturity has been assessed using a variety of methods, including analysis of skeletal age, pubertal stage, percentage of adult height, and age at peak velocity (Malina et al., 2015)”. It is strongly suggested that you clarify why these methods are invasive.
REPLY: This is now clarified.
- Several paragraphs without bibliographic references; it is strongly suggested that references from the last 5 years be included.
REPLY: We have now added in citations.
- Clearly describe the hypothesis.
REPLY: We have now made the aims of the study clearer in the last paragraph of the Introduction.
- Write the justification for the study and its contributions to the learning process and performance of young footballers.
REPLY: We have now added in the text:
“This offers the opportunity for relatively young or late-maturing players to be more objectively assessed in terms of their talent potential.”
- It is strongly recommended to describe the aim of the study.
REPLY: No clearly defined.
Methods
- Describe the study design.
- Describe the setting, locations, and relevant dates, including periods of recruitment, exposure, follow-up, and data collection. In this sequence.
- Describe the participants: Give the eligibility criteria, and the sources and methods of selection of participants
REPLY: All corrected
- Insert the mean and standard deviation in scientific form ( m±dp) and insert in the results section, not in the method.
REPLY: All corrected
- Describe specifically how the protocol was measured, what instruments were used?
REPLY: All instrumentation added.
- It is strongly suggested to draw up an experimental design of the protocol in animation format to better clarify the protocol.
REPLY: There is a figure used in previous papers to demonstrate the design of the dribbling circuit. We could add this in here but seems unnecessary. We have provided an appropriate citation for this, but we are happy to add such a figure if deemed necessary.
- Lack of references for the protocols; if the authors have developed them, base them on other articles using the term “adapted from (authors et al., 20XX).
REPLY: We have made this much clearer in the Methods.
- Describe any efforts to address potential sources of bias
REPLY: We have now noted these points.
- Table 1. It should be corrected. In the header, only the title of the table, the tests used, and the legend should be inserted at the bottom of the table. Keep the standard in all tables. The mean and standard deviation need to be formatted according to scientific standards.
- There are many tables and data, it is suggested to select the most relevant information and present it.
REPLY: We have now made these corrections.
Results
We suggest revising the English language of the tables and figures, as well as throughout the manuscript.
- REPLY: We have gone through the entire manuscript, as suggested.
Discussion and Conclusion
- The text presents the results in a scattered way. Be clearer and more objective in your inferences.
REPLY: We have now added these points.
Reviewer 2 Report
Comments and Suggestions for Authors
The article aims to: we assessed how a player’s sprinting performance, directed dribbling, and age and size index (age, height, and mass), affects their overall dribbling performance.
The authors of the research are requested to address the following items:
1) The use of the verb "Quantify" to describe the research objective in the "Abstract" section (Line: 2) is insufficient. According to the research results, the verb level should be higher. Example: "Demonstrate."
2) Clearly state the research objective in the final paragraph of the introduction.
3) In the "Methods" section, describe the type of research conducted.
4) It is unclear whether the studied sample (180 football players) is representative of the population or at least sufficient to establish strong correlations. In either case, describe the sampling method and the statistical measures required to meet the assumptions of representativeness, statistical power, or effect size to determine sample sufficiency.
5) In the "Statistical Analysis" subsection, it is unknown whether the data distribution follows normality. Specify the normality test used, which should justify the choice of the correlational statistic applied.
6) It is recommended to use the exact same research objective stated at the beginning of the study (to be corrected in the Discussion section).
7) The limitations of the study should be included as part of the Discussion section.
8) A Final Considerations section is requested, summarizing the key findings of the research to provide the reader with a concise overview.
Author Response
Thank you for your review -I am replied to each of your comments and queries below.
Reviewer 2
The authors of the research are requested to address the following items:
The use of the verb "Quantify" to describe the research objective in the "Abstract" section (Line: 2) is insufficient. According to the research results, the verb level should be higher. Example: "Demonstrate."
REPLY: Just in this specific case the word “quantify” is much more specific to our intended meaning than “demonstrate” - but we have clarified in other places with this suggestion in mind. Thank you.
- Clearly state the research objective in the final paragraph of the introduction.
REPLY: Thank you – yes, we have now tried to be more clear and direct here.
- In the "Methods" section, describe the type of research conducted.
REPLY: We have now extensively edited the methods.
- It is unclear whether the studied sample (180 football players) is representative of the population or at least sufficient to establish strong correlations. In either case, describe the sampling method and the statistical measures required to meet the assumptions of representativeness, statistical power, or effect size to determine sample sufficiency.
REPLY: This is an excellent point. We have modified the text to make it clear that all of the players in this elite Brazilian football academy were sampled. Many of these players have now gone on to become professional players.
- In the "Statistical Analysis" subsection, it is unknown whether the data distribution follows normality. Specify the normality test used, which should justify the choice of the correlational statistic applied.
REPLY: We have now clarified.
- It is recommended to use the exact same research objective stated at the beginning of the study (to be corrected in the Discussion section).
REPLY: Thank you – yes, we have made this adjustment acoordingly.
- The limitations of the study should be included as part of the Discussion section.
REPLY: We have now added some limitations
- A Final Considerations section is requested, summarizing the key findings of the research to provide the reader with a concise overview.
REPLY: We have now added these points.
Reviewer 3 Report
Comments and Suggestions for Authors
Suggestions for changes:
- The objective should be specified in more detail, as the reader might be left somewhat uncertain.
- Since minor soccer players were included in the study, it would be advisable to mention how parental consent was obtained for their participation.
- The advantages of the study should be clearly stated.
- It would be useful to include the limitations of the research.
- Finally, implications for future research and practical contributions to the training process of young soccer players should be outlined.
Finally, I believe this article is acceptable for publication in the journal, and my suggestion is to accept the manuscript with minor corrections

Author Response
Reviewer 3
REPLY: Thank you for your generous and complementary comments on our manuscript. It is very much appreciated.
Suggestions for changes:
- The objective should be specified in more detail, as the reader might be left somewhat uncertain.
- Since minor soccer players were included in the study, it would be advisable to mention how parental consent was obtained for their participation.
- The advantages of the study should be clearly stated.
- It would be useful to include the limitations of the research.
- Finally, implications for future research and practical contributions to the training process of young soccer players should be outlined.
REPLY: We have now gone through the entire paper and adjusted according to all these points and suggestions.
Reviewer 4 Report
Comments and Suggestions for Authors
The introduction section is too long and should be shortened.
…Gibson’s theory of affordances (Gibson, 1966, 1977, 1986). Isn’t this theory quite old? It might be easier to refute.
Method: The term guardian can be replaced with parent or legal guardian.
The division of study groups is appropriate (U12, … U20).
The statistical method used is correct.
The discussion section is too superficial. The most recent reference used is Hunter et al., 2021. References from the last three years should be added, focusing on cause-and-effect relationships.
What are the limitations of the study? They should be stated.
Table 1: The letter "N" is written in uppercase. An uppercase "N" refers to the population, while a lowercase "n" refers to the sample. Therefore, a lowercase "n" should be used.
The topic of the study is well-chosen, and it yields impactful results. Implementing the suggested revisions will enhance the study's overall impact.
Author Response
Reviewer 4
The introduction section is too long and should be shortened.
…Gibson’s theory of affordances (Gibson, 1966, 1977, 1986). Isn’t this theory quite old? It might be easier to refute.
REPLY: This is still an influential idea in Cognitive psychology.
Method: The term guardian can be replaced with parent or legal guardian.
REPLY: Corrected.
The division of study groups is appropriate (U12, … U20).
The statistical method used is correct.
REPLY: Thank you.
The discussion section is too superficial. The most recent reference used is Hunter et al., 2021. References from the last three years should be added, focusing on cause-and-effect relationships.
REPLY: We have gone through the discussion with these points in mind and added references where appropriate. Thank you.
What are the limitations of the study? They should be stated.
REPLY: We have now added some limitations
Table 1: The letter "N" is written in uppercase. An uppercase "N" refers to the population, while a lowercase "n" refers to the sample. Therefore, a lowercase "n" should be used.
REPLY: We have made these changes
The topic of the study is well-chosen, and it yields impactful results. Implementing the suggested revisions will enhance the study's overall impact.
REPLY: The comments and suggestions are much appreciated
Round 2
Reviewer 1 Report
Comments and Suggestions for Authors
Dear colleagues, here are the contributions to improve the manuscript.
Yours sincerely

The manuscript still needs professional proofreading in English.
Author Response
Reviewer 1
Thank you again for your review and suggestions. As you will see from the tracked changes within the document, I have made considerable changes to the manuscript to better reflect your suggestions.
The following are suggestions for adjustments and possible issues to be considered by the authors:
Title:
Recommendation: I would also urge the authors to review the title of the article, does the term “size” refer to height or body composition?
REPLY: Thank you for the clarification question. We have used the term “body size” because it does not refer to body composition but the two traits of height and mass. Thus, the term most appropriate in this case would be “body size”.
Abstract
Please include the study design in the abstracts.
REPLY: We hope the following text satisfies the description required in the Abstract:
“Background/Objectives: Dribbling is a fundamental skill in soccer but assessing the performance of youth players in this skill is complicated by the confounded effects of age and physical development. In this study, our aim was to quantify the interactive effects of age, height, and mass on the dribbling performance of 180 players between 10 and 21 years old from an elite Brazilian junior academy. Methods: For each player, we quantified their dribbling and sprinting speed along four different paths with varying curvature, and their ability to perform specific, directed dribbling drills using one or both feet. To characterise patterns of variation among player’s age, height, and mass – and to control for their confounding effects - we used a principal component analysis (PCA) to create a multivariate index of age and size (ASI).”
Introduction
We strongly suggest updating the references used in the introduction, looking for articles from the last 5 years (there are still many references that are more than 5 years old).
REPLY: We have now added many more recent references.
Methods
Describe the study design and present the bioethical issues (“without this description there is no way to ‘publish the manuscript’”).
REPLY: The following text has been modified accordingly:
Lines 154-161: “We measured the dribbling and sprinting performance of all 180 individuals across seven teams (U12, U13, U14, U15, U16, U17, U20) from a Tier 1 professional academy that competes in their state and national competitions in Brazil. For each individual, we also recorded their date of birth, mass (± 0.1kg; Ulysses, USA) and standing height (± 0.01m). All players and parental and legal guardians gave consent to be involved in the study, which was in accordance with ethical protocols for the University of Queensland, Australia, and State University of Londrina, Brazil (#2019001398). All data were analysed anonymously. No activities or movements beyond those regularly utilised by the players in their training sessions were involved in the study design.”
Results
No comments or notes.
Discussion and Conclusion
No comments or notes.
References
It is also strongly suggested to replace references that are more than 5 years old in the introduction and final references.
REPLY: See above.
Reviewer 2 Report
Comments and Suggestions for Authors
- Point 4 has not been resolved.
- Point 4 has not been resolved.
- Point 5 has not been resolved.
- Point 7 has not been resolved.
It is recommended to specify the page numbers where the requested changes have been made in future submissions.
Author Response
Response to Reviewer 2:
Apologies for the miscommunication in our first reply. Below we have answered each point raised more clearly and we have highlighted specific sections that did (somewhat) address your original queries but where I did not clearly highlight the changes. Thank you again for your time and consideration. As requested, we have readdressed Points 4, 5 and 7.
Point 4
- It is unclear whether the studied sample (180 football players) is representative of the population or at least sufficient to establish strong correlations. In either case, describe the sampling method and the statistical measures required to meet the assumptions of representativeness, statistical power, or effect size to determine sample sufficiency.
REPLY: I have now added the following text to the manuscript:
Lines 319-322 “We measured the dribbling and sprinting performance of all 180 individuals across seven teams (U12, U13, U14, U15, U16, U17, U20) from a Tier 1 professional academy in Brazil that compete in their state and national competitions.”
Lines 483-487:” “We used power analyses to determine if we had sufficient statistical power to detect significant effects for both continuous traits (regression) and among groups (ANOVA) (R Core Team, 2020). We assumed a medium effect size in each case, an alpha level of 0.05, and power of 0.80. Results indicated we had sufficient statistical power across all analyses.”
We have also added the following text to the Discussion to discuss the limitations of the statistical design based on only one club.
Lines 674-679: “In this study, we provide normative data that can allow one to assess the soccer-specific skill of youth soccer players after being adjusted for age and size. With the normative equations provided, one can more equitably compare players’ dribbling performances. However, one should be mindful that these equations are based on athletes from only one group of elite youth players and it remains to be tested how generalizable these are to players from other clubs, countries or continents.”
Point 5
In the "Statistical Analysis" subsection, it is unknown whether the data distribution follows normality. Specify the normality test used, which should justify the choice of the correlational statistic applied.
REPLY:
Lines 483-484: “Normality of residuals was assessed using the Shapiro-Wilk test and visual inspections of Q-Q plots using R.”
Point 7
The limitations of the study should be included as part of the Discussion section.
We have added to the following sections:
Lines 622-659:
“There is little doubt that possessing close ball control in isolated conditions will be a pre-requisite for match-realistic dribbling ability against opponents, but there is no guarantee that training-induced increases in technical dribbling will directly map onto attacking performance against opponents. This is a key limitation of our study. Dribbling speeds, as measured along the curved paths, are only correlated with an individual’s attacking and defensive ability in 1v1 competitions (Wilson et al., 2018, 2019, 2020) and we do not know the causal relationship between our metric of dribbling and the actual activity utilised in game-realistic situations. Our measure of dribbling performance relies on the execution of closed-skills, which have often been criticised for de-contextualising skills from the performance setting (Bergkamp et al., 2019; Phillips et al., 2010; Williams & Hodges, 2008). However, we do expect improvements in dribbling speed along the testing paths will be associated with improvements in a player’s attacking and defensive capabilities during match-realistic conditions, but this still remains to be tested.”
Lines 674-681:
“In this study, we provide normative data that can allow one to assess the soccer-specific skill of youth soccer players after being adjusted for age and size. With the normative equations provided, one can more equitably compare players’ dribbling performances. However, one should be mindful that these equations are based on athletes from only one group of elite youth players and it remains to be tested how generalizable these are to players from other clubs, countries or continents. Regardless, the selection of talented players should be rapidly moving towards the inclusion of quantitative metrics that help minimise biases of age and size – such as the metrics described in this study – to ensure both selection fairness and accuracy in youth team sports.”
Round 3
Reviewer 2 Report
Comments and Suggestions for Authors
Most of the issues have been resolved.